# Fractional Calculus Approach to Reproduce Material Viscoelastic Behavior, including the Time–Temperature Superposition Phenomenon

**DOI:** 10.3390/polym14204412

**Published:** 2022-10-19

**Authors:** Andrea Genovese, Flavio Farroni, Aleksandr Sakhnevych

**Affiliations:** Department of Industrial Engineering, University of Naples Federico II, Via Claudio 21, 80125 Naples, NA, Italy

**Keywords:** viscoelasticity, material parametrization, WLF coefficients, pole–zero formulation, fractional model

## Abstract

The design of modern products and processes cannot prescind from the usage of viscoelastic materials that provide extreme design freedoms at relatively low cost. Correct and reliable modeling of these materials allows effective use that involves the design, maintenance, and monitoring phase and the possibility of reuse and recycling. Fractional models are becoming more and more popular in the reproduction of viscoelastic phenomena because of their capability to describe the behavior of such materials using a limited number of parameters with an acceptable accuracy over a vast range of excitation frequencies. A particularly reliable model parametrization procedure, using the poles–zeros formulation, allows researchers to considerably reduce the computational cost of the calibration process and avoid convergence issues typically occurring for rheological models. The aim of the presented work is to demonstrate that the poles–zeros identification methodology can be employed not only to identify the viscoelastic master curves but also the material parameters characterizing the time–temperature superposition phenomenon. The proposed technique, starting from the data concerning the isothermal experimental curves, makes use of the fractional derivative generalized model to reconstruct the master curves in the frequency domain and correctly identify the coefficients of the WLF function. To validate the methodology, three different viscoelastic materials have been employed, highlighting the potential of the material parameters’ global identification. Furthermore, the paper points out a further possibility to employ only a limited number of the experimental curves to feed the identification methodology and predict the complete viscoelastic material behavior.

## 1. Introduction

With the advancements in modern technology, the continuous evolution of materials, and more efficient manufacturing processes, the design of modern products and processes cannot prescind from the usage of viscoelastic materials. These kinds of materials provide extreme design freedoms at relatively low cost: high elasticity and impermeability, adequate chemical and heat resistance, insulation, and an ability to absorb shocks and dampen noise [1,2]. A holistic approach to knowledge-based material selection allows researchers to not only adequately select materials from the perspective of functional attributes but also consider their emotional and esthetic qualities, contemplating the added sustainable value, such as recyclability, energy efficiency, and solar-power capacity [3,4].

The knowledge of the material and the ability to properly model the material characteristics starting from the earliest design stages becomes mandatory in cases where the aim is to govern the material properties through the manufacturing processes and for the entire product lifecycle, taking into account its eventual changes during aging to overcome the typical limits and constraints in a design path where materials and transformation technologies are both variables of the creation process [5,6]. The crucial aspects in this context concern the choice of the proper mathematical formulation for the material modeling and the ability to calibrate the model with a limited amount of data explored in a particularly narrow frequency or temperature range, whose accuracy can be affected by the experimental technique employed. Indeed, the aim of both destructive and non-destructive testing techniques is to exploit the widest possible frequency–temperature working domain of the material to understand the material behavior within the exploited operating conditions and provide a sufficient amount of data for model-calibration purposes [7,8,9,10].

The constitutive law for linear viscoelasticity, based on Boltzmann’s superposition principle [3], can be established by means of three different approaches: integral models, linear differential models, and fractional derivative models. The differential approach is used to describe the rheological properties by means of linear differential equations that link stress and strain [11]. A combination of mechanical elements, ideal springs, and dashpots are used to build suitable rheological models. The generalized models are often employed to describe the viscoelastic behavior of the materials in a wide range of frequencies and time scales. If, on the one hand, these models offer a good description of the viscoelastic materials, on the other hand, they involve a set of differential equations to describe the dynamic state of the system, which could considerably complicate the overall mathematical formulation, significantly increasing the computational load due to a larger set of motion equations to be solved. To overcome these issues, fractional models are becoming more and more popular because of their ability to reproduce the behavior of viscoelastic materials using a limited number of parameters with an acceptable accuracy level over a vast range of excitation frequencies, combining ideal spring and spring-pot elements [12].

Several applications take advantage of fractional models. A review regarding the application of fractional calculus in the models of linear viscoelasticity utilized in dynamic problems of mechanics of solids has been conducted by Shitikova [13]. Abouelregal [14] proposed a methodology to study thermoelastic vibrations in a homogeneous isotropic three-dimensional solid based on a fractional derivative Kelvin–Voigt model. In [15], Zhou et al. adopted a variable-order fractional derivative material model to numerically analyze the behavior of the frozen soil, including creep, stress relaxation, and strain rate effects. In [16], Wang et al. adopted the fractional derivative model to describe the hysteretic behavior of the magnetorheological elastomers, demonstrating a huge potential in the field of intelligent structures and devices.

Furthermore, in [2], the authors presented a particularly reliable procedure adopting the poles–zeros formulation, which considerably reduced the calculation effort of the identification/calibration process. Starting from these results, the aim of this work is to demonstrate that it is also possible to identify both the master curve and the WLF function by adopting the poles–zeros identification methodology. The technique involves the individual isothermal experimental curves and makes use of the fractional derivative generalized model poles–zeros identification to reconstruct the master curve and correctly identify the coefficients of the WLF function. Furthermore, this paper points out a further possibility to employ only a limited number of experimental isothermal curves to feed the proposed identification methodology (such as the one addressing the material glass transition zone). It has to be highlighted that at the current stage, the vertical shifts have been neglected (this hypothesis is fine if the experimental acquisitions do not cover particularly cold temperatures or, on the other side/dually, particularly high testing frequencies), but they could also be part of the parameters to be identified in further investigations.

The paper is organized as follows: in Section 2, the fractional derivative generalized model is defined in the frequency domain to obtain the relative poles–zeros formulation; in Section 3, the WLF time–temperature superposition principle is defined with a particular focus on the determination of the horizontal and vertical shifts and an analytic study to demonstrate the WLF validity in the poles–zeros domain; in Section 4, the possibility of adopting the fractional models and the poles–zeros formulation to determine not only the viscoelastic moduli but also the WLF shift factor starting from the experimental data is investigated; in Section 5, a specific study to understand how the minimum number of experiments affects the identification procedure is presented.

## 2. Fractional Derivative Model in Poles–Zeros Formulation

Viscoelastic materials, due to their intrinsic rheological stress–strain dependence on time, which is based on the fact that the deformation energy is not totally stored but partially dissipated through a hysteretic mechanism, exhibit both elastic and viscous characteristics concurrently [17]. To analyze the mechanical response of viscoelastic materials, three different testing methodologies are usually adopted: static, transient, and dynamic [18]. Whereas the static characterization regards the static or quasi-static application of load or deformation, transient (creep and stress relaxation experiments) and dynamic (DMA and DSC techniques) testing procedures concern the analysis of material response towards time once deformation or load functions (elongation or shear) are applied [10,19,20]. The dynamic behavior of this kind of material is characterized by the material dynamic stiffness *E**, which is a complex variable defined as (1):(1)$$\frac{\sigma \left(\omega \right)}{\varepsilon \left(\omega \right)}=E^{*}=E^{\prime }+E^{\prime \prime }$$
where *E*′ is the storage modulus (Pa), E′′ is the loss modulus (Pa), σ(ω) is the cyclic stress applied to the material (Pa) at given circular frequency ω, and ε(ω) is the corresponding strain response.

These quantities are linked to the way the material dissipates energy provided by means of a load/stress time function. Particularly, they are related to the phase angle δ, according to Equation (2):(2)$$\frac{E^{\prime \prime }\left(\omega \right)}{E^{\prime }\left(\omega \right)}=tan\delta$$

It is worth noting that all the physical quantities, referred to as the properties of the viscoelastic material’s behavior, are a function of the particular frequency levels at which sinusoidal load/deformation is applied during the test. More precisely, the modulus, the energy loss, and the hysteresis of a viscoelastic material change in relation to two parameters: the frequency the force is applied with and the temperature at which the phenomena are evaluated.

The correct modeling of the material viscoelastic properties is a key element in achieving reliable results from analytical models’ outputs or finite-element-based analyses within the design of the desired dynamic behavior of mechanical systems. Several mathematical models can be found in the literature to help understand and describe material viscoelastic behavior [21]. The Maxwell and Kelvin models fail to represent the actual response of viscoelastic materials at low and high frequencies, respectively, while generalized models return more accurate results but imply a more complicated mathematical formulation and increase computational costs. As this paper is focused on viscoelastic solids, the Fractional Derivative Generalized model (FDGM) is adopted by the authors to model the viscoelastic behavior. This model, depicted in Figure 1, is obtained by connecting fractional Maxwell cells in series, where a fractional Maxwell cell is defined as a spring and a spring-pot element arranged in series.

From a mathematical point of view, a generic constitutive equation for viscoelastic materials, based on fractional derivative orders, is expressed in the following Equation (3):(3)$$\overset{N}{\underset{n=0}{\sum }}a_{n}\frac{d^{\alpha_{n}}\sigma \left(t\right)}{dt^{\alpha_{n}}}=\overset{M}{\underset{m=0}{\sum }}b_{m}\frac{d^{\beta_{m}}\varepsilon \left(t\right)}{dt^{\beta_{m}}}\;\;$$
where $\alpha_{n}$ and $\beta_{m}$ are the fractional derivative orders included within the range [0, 1], N=M and $b_{0}=0$ for the considered Maxwell formulation. Turning to the frequency domain by applying the Fourier transform and assuming that α=β, Equation (3) gives the following expression for the complex Moduli (4):(4)$$E^{*}\left(i\omega \right)=E_{0}+\overset{N}{\underset{k=1}{\sum }}\frac{{\left(i\omega \right)}^{\alpha_{k}}E_{k}\eta_{k}\;}{E_{k}+{\left(i\omega \right)}^{\alpha_{k}}\eta_{k}}$$
where the parameters $E_{k}$ and $\eta_{k}$ represent the springs’ stiffness and spring-pot coefficient, respectively, as represented in Figure 1, and ω is the angular frequency.

Renaud et al. [22] have shown that, in analogy with the generalized models, Expression (4) can be equivalently expressed in the poles–zeros formulation as (5):(5)E*(iω)=E0 ∏k=1N1+(iωωz,k)αk1+(iωωp,k)αk
where $\omega_{z}$ and $\omega_{p}$ are the zeros and poles, respectively. The presented approach, consisting of a superposition of poles–zeros coupled behavior, overcomes the computational and convergence issues typically occurring in the parameterization of the rheological models in the time and frequency domain forms since it enables the determination of boundary conditions and initial starting guess for the optimization problem [23]. Figure 2 schematizes the main phases of the adopted constrained nonlinear optimization procedure aiming at identifying a robust set of poles–zeros coefficients, starting from a feasible set of initial conditions, detailed by the authors in [2].

## 3. Time–Temperature Superposition

Due to the limitation of most commercial DMA equipment, the tests for the determination of the complex modulus are usually limited to the low-frequency range domain [24]. To extend the material properties to a broader frequency range, which is necessary for applications such as contact, friction, and wear modeling approaches, the time–temperature superposition (TTS) is applied [25]. Time–temperature superposition (also called frequency–temperature superposition or method of reduced variables) is a frequently applied procedure to determine the temperature dependence of the rheological behavior of a polymer and to expand the time or frequency regime at a given temperature at which the material behavior is studied.

As observed in [26,27], the frequency–temperature superposition is valid due to the fact that various relaxation times belonging to a given relaxation process have the same temperature dependence. This principle relates the material response at a given time *t* (or frequency *ω*), and at a given temperature *T*, to that at other conditions (denoted by subscript *r*):(6)$$\omega_{r}=a_{T}\left(T,T_{r}\right)\;\omega \;E^{\prime }\left(\omega_{r},\;T_{r}\right)=b_{T}\left(T,T_{r}\right)E^{\prime }\left(\omega ,T\right)\;E^{\prime \prime }\left(\omega_{r},\;T_{r}\right)=b_{T}\left(T,T_{r}\right)E^{\prime \prime }\left(\omega ,T\right)$$
where *a_T_* (*T*, *T_r_*) and *b_T_* (*T*, *T_r_*) are coefficients that indicate the amount of horizontal and vertical shifting (respectively) to be applied to isotherms of storage and loss moduli measured at a temperature *T* in order to estimate the material properties at a reference temperature *T_r_*, as qualitatively shown in Figure 3 for explanatory purposes.

The horizontal shift factors *a_T_* (*T*, *T_r_*) describe the temperature dependence of the relaxation time and usually follow the empirical Williams–Landel–Ferry (WLF) law (7):(7)$$\log_{10}a_{T}\left(T,T_{r}\right)=-\frac{C_{1}\left(T-T_{r}\right)}{C_{2}+\left(T-T_{r}\right)}$$
where *C*_1_ and *C*_2_ are empirical constants whose order of magnitude is about 10 and 100 K, respectively.

The vertical shift factors *b_T_* (*T*, *T_r_*) are related to thermal expansion effects, which for most polymers can be neglected due to their small variation, and that, for this reason, will be neglected in the following. Here, it is worth noting that this hypothesis is highly acceptable in the viscoelastic regions where the frequency/time dependence of material functions is sharp. On the other hand, overlooking thermal vertical shifts in viscoelastic regions with weak frequency/time dependence may lead to different values of horizontal shifting whose accuracy depends on the material under investigation.

Figure 4 shows the typical qualitative trend of the vertical and horizontal shift factors as a function of the difference from a reference temperature *T_R_*.

The question arises regarding the eventual validity of the WLF principle through the material transformation to the poles–zeros formulation. To this end, the validity of the WLF principle has been investigated starting from a completely characterized polymeric material, whose characteristics in terms of the experimental curves (*E′* and *tan(δ)*) and parameters in terms of WLF coefficients (*C*_1_ and *C*_2_) have been measured and calculated, respectively. To transform the FDGM in poles–zeros formulation for two different temperatures, *T_r_* an *T*_1_, the standard procedure consists of the following steps:Identification of poles and zeros at a reference temperature *T_r_*, applying the scheme represented in Figure 2;Application of the WLF formulation to the experimental curves referring to a reference temperature *T_r_* to obtain the experimental curves at the new temperature *T*_1_, shifting all frequency vector *ω_exp_*;Identification of the poles and zeros starting from the experimental curves referring to the new temperature *T*_1_ by means of the procedure summarized in Figure 2.

It should be highlighted that this procedure is computationally consuming since the identification procedure must be performed as many times as the temperatures of interest.

In this study, the authors also investigate an alternative procedure to perform the shift of the curves:Identification of the poles and zeros at reference temperature *T_r_*, with the procedure described in Figure 2;Application of the WLF law directly on the identified poles and zeros, obtaining the master curves at the new temperature *T*_1_.

The two procedures, validated by a significant amount of experimental data, provide the exact same results, which can be mathematically summarized as follows (8):(8)$$\begin{array}{c} {E^{\prime }}_{T1}=f\left(\frac{\omega_{exp}\left(T_{r}\right)}{a_{T}};\omega_{p,k}\left(T_{1}\right);\omega_{z,k}\left(T_{1}\right);\alpha_{k}\right)=f\left(\omega_{exp},\frac{\omega_{p,k}\;\left(T_{r}\right)}{a_{T}},\frac{\omega_{z,k}\;\left(T_{r}\right)}{a_{T}},\alpha_{k}\right)\\ tan\delta_{T1}=f\left(\frac{\omega_{exp}\left(T_{r}\right)}{a_{T}};\omega_{p,k}\;\left(T_{1}\right);\omega_{z,k}\left(T_{1}\right),\alpha_{k}\right)=f\left(\omega_{exp},\frac{\omega_{p,k}\;\left(T_{r}\right)}{a_{T}},\frac{\omega_{z,k}\;\left(T_{r}\right)}{a_{T}},\alpha_{k}\right) \end{array}$$
where $\omega_{exp}$ is the frequency of the experimental test.

Therefore, it is possible to generalize the poles–zeros Formulation (5) by including the WLF (9):(9)$$E^{*}\left(i\omega \right)=E_{0}\;\overset{N}{\underset{k=1}{\prod }}\frac{1+{\left(\frac{i\omega }{\left(\frac{\omega_{z,k}}{a_{T}}\right)}\right)}^{\alpha_{k}}}{1+{\left(\frac{i\omega }{\left(\frac{\omega_{p,k}}{a_{T}}\right)}\right)}^{\alpha_{k}}}$$
where *a_T_ =* 1 when *T = T_rif_*.

From Equations (8) and (9), indicating with $\omega_{p,k_{rif}}$ and $\omega_{z,k_{rif}}$, respectively, the poles and zeros identified at the reference temperature *T_rif_*, it is possible to write the WLF function applied to the poles–zeros formulation as (10):(10)$$\frac{\omega_{p,k}\left(T\right)}{\omega_{p,k_{rif}}}=\frac{\omega_{z,k}\left(T\right)}{\omega_{z,k_{rif}}}=a_{T}\left(T-T_{rif}\right)=10^{-\frac{C_{1}\cdot \left(T-T_{rif}\right)}{C_{2}+\left(T-Trif\right)}}$$

The equivalence between the Equations (7) and (10) is highlighted in Figure 5 for a polymer of known parameters *C*_1_ and *C*_2_ and adopting FDGM with three elements (N = 3). It should be noted that for an FDGM with three elements, the subscript *k* in (10) is three. This means that, for each temperature, three poles and three zeros are identified, and each of them satisfies Equation (10).

Therefore, the validity of the WLF principle through the material transformation to the poles–zeros formulation has been highlighted. Here, it is worth noting that all that has been demonstrated for the FDGM pole–zeros formulation is still valid for the generalized formulation (in Maxwell and Kelvin–Voight form).

## 4. Material Parameters’ Global Identification

### 4.1. Identification Procedure

Since Equation (10) demonstrated the relation between the WLF function and the poles and zeros, the authors aimed to study the possibility of adopting the fractional models and the poles–zeros formulation to determine not only the viscoelastic moduli but also the WLF shift factor starting from the experimental data. With this purpose in mind, the authors, starting from the experimental data obtained by the DMA, make use of the FDGM to obtain the viscoelastic mater curves. Figure 6 shows, for a polymer compound named A, the curves obtained by the experimental DMA, consisting of a series of isotherms in a frequency range of 0–100 rad/s. 

To build the moduli master curves as function of the frequency at one temperature, an arbitrary reference temperature must be set. In order to minimize the randomness of the procedure, a unique way to establish the reference temperature has been defined. The following it has been assumed that T_rif_ is the temperature of the isotherm curve that presents the maximum value of the tan δ (i.e., for compound A, *T_rif_* = −20 °C). 

The number of parameters to be identified is 3N + 3:3N + 1 parameters for the pole, zero, and static modulus (9);Two parameters for the WLF (10).

Since in previous work, it has been shown that three is a sufficient number to obtain an excellent fitting [2], an FDGM with N = 3 fractional Maxwell cells has been considered in the following. According to the procedure defined in [2], the parameters’ identification is performed by means of a constrained procedure, seeking the minimum of a nonlinear error function *W* (11): of the 3N + 3 real variables: (11)$$W=\left(1-r\right)\cdot \;Err_{E}+r\cdot \;Err_{tan\delta }$$
where $Err_{tan\delta }$ and $Err_{E}$ are defined by (12) and (13), and r is the weight factor:(12)$$Err_{E}=\frac{\sqrt{\frac{\sum_{i=1}^{N}{\left(E-E_{model}\right)}^{2}}{N}}}{mean\left(E\right)}$$
(13)$$Err_{tan\delta }=\frac{\sqrt{\frac{\sum_{i=1}^{N}{\left(tan\delta -tan\delta_{model}\right)}^{2}}{N}}}{mean\left(tan\delta \right)}$$

In other words, Error Function (11) is defined as the weighted sum of the normalized root mean square error (NRMSE) calculated for the storage modulus and the loss tangent as in (12) and (13). It is worth noting that an error function so defined allows fitting modulus, phase, and WLF coefficient at the same time. The weighting coefficient *r* has been assumed to be equal to 0.5 for all in the following to test the procedure in the same conditions.

Regarding the initial condition of the identification procedure, the initialization of the pole–zero parameters has been carried out by means of the method proposed by Renaud et al. [22]; while for initial *C_1_* and *C_2_* in (10), they have been assumed to be 17.44 and 51.60 respectively, which are generally accepted values when *T_rif_* = T_g_ [26].

### 4.2. Results

The identification procedure described in the previous sub-section has been employed by the authors for three different polymers’ compounds, A, B, and C, to evaluate the capability of the FDGM models to describe three completely different viscoelastic materials in terms of time–temperature superposition. 

The thermal properties of samples were investigated by using a TA DSCQ2000 differential scanning calorimeter equipped with a TA Instruments DSC cooling system. Dry nitrogen gas with a flow rate of 20 mL/min was purged through the cell during the measurements and the thermal treatments. Samples of approximately 8 mg were heated from −80 to 100 °C and kept at this temperature for 3 min, then cooled from 100 to −80 °C at 50 °C/min, kept at this temperature for 3 min, and re-heated from −80 to 100 °C at 20 °C/min. The heating rate was fixed to 20 °C/min, whereas the cooling was carried out at 50 °C/min. 

In particular, the heat flows for different compounds are represented in Figure 7, and the identified glass transition temperatures (Tg), evaluated in the second heating run, are listed in Table 1.

All the samples exhibited a weak glass transition temperature around 9 °C, while only samples B and C showed an intense transition at −33 and −20 °C, respectively. 

For compound A, Figure 8a compares the experimental master curves with the results obtained by the identification procedure, while in Figure 8b, the WLF law is shown. 

From a qualitative point of view, the FDGM with three fractional elements is able to provide an acceptable representation of the master curves’ shapes and the WLF law. Particularly in Figure 8b, it is possible to note that for temperatures close to *T_rif_* = −20° (*T* − *T*_0_ = 0), the procedure is quite accurate, while for temperatures far from this value, some differences appear from the experimental data. These small differences (<10%) are due to the vertical shift factors *b_T_* (*T*, *T_r_*) that, in this work, were neglected. Figure 9a,b and Figure 10a,b show the results for the compound B and C, respectively. For both compounds, *T_rif_* = −20°.

To quantify the accuracy of the FDGM approach, the NRMSE is also evaluated, both for the storage modulus and for the loss tangent quantities defined in Equations (12) and (13). The purpose of these indicators is to quantify the goodness of the models’ behavior toward the experimental data reproduction. Table 2, in addition to the NRMSE, reports the identified coefficients *C_1_* and *C_2_* of the WLF law (10).

The identified parameters of these models are detailed in Table A1 in Section A.1. Analyzing the results, it is possible to note that the FDGM returns a very good fitting of experimental data with an NRMSE < 0.05 for all the compounds. 

Regarding the WLF coefficients, the procedure is able to identify three different couples of parameters, one for each polymer compound. We can note that each identified couple is different from the starting set.

## 5. Mater Curves and Time–Temperature Superposition Parameters’ Estimation with Partial Experimental Data

In the previous sections, it has been demonstrated how the fractional calculus in poles–zeros formulation is a powerful analytical model able to reproduce the material viscoelastic behavior, including the time–temperature superposition phenomenon. The FDGM ensures not only a high correlation with the viscoelastic master curves but also is able to estimate the different behavior of each polymer compound in terms of time–temperature superposition, returning different WLF laws for each of them. Taking advantage of this methodology, the question arises about the possibility of reproducing the viscoelastic behavior with a minimum number of experimental data. The aim of the following analysis is to test the capability of the FDGM model in moduli and WLF coefficients estimation, starting from a reduced set of experimental isotherms from DMA data on which to make the fitting. It is worth noting that a reduction in the experimental test necessary to characterize the viscoelastic behavior would lead to savings in resources, money, and time.

In [2], it has been demonstrated that an excellent prediction of the master curves can be achieved by adopting data coming from the lower and upper frequencies plateau of the storage modulus, the peak of the loss tangent curve, and the curvature change of both curves. Moreover, satisfying results can also be obtained by considering only the data available of the upper and low-frequency plateaus plus those at the loss tangent peak. Borrowing these results, three isotherms have been chosen for the analysis: an isotherm at high temperature, an isotherm at low temperature, and one that presents the maximum value of tan δ. For explanatory purposes, Figure 10 reports the experimental starting set for the identification of compound C.

The parameters of the three-element FDGM and of the WLF law have been identified, and the models’ results have been compared with the experimental results in the entire frequency range. Figure 11 depicts the curve comparisons for compound C: Figure 11a shows the master curves estimated (FDGM reduced) compared with the full experimental DMA. Furthermore, in the same figure, the partial experimental data used for the identification are highlighted in the frequency domain. Figure 11b shows the experimental WLF law in comparison with the identified value of the FDGM by adopting both all and partial experimental data. These results highlight how the proposed technique is able to give a good estimation of the master curves and the time–temperature superposition parameters, while also using a reduced dataset of experimental data. To quantify the approximation in the latter case, the NRMSE has been calculated in the entire frequency range and for the three compounds. The results are reported in Table 3, where it is possible to note that the NRMSE < 0.06 in all cases. Some differences can be found in the estimation of parameters *C_1_* and *C_2_*; however, this difference implies a small difference in the calculation of *a_T_* in the temperature range exanimated, as shown in Figure 12a,b.

The identified parameters for the three compounds, A, B, and C, using partial experimental data, are detailed in Table A2 in Appendix A.

## 6. Conclusions 

The search for rheological models to be used for the design and study of components made of viscoelastic material is a topic of great interest both from the industrial and the academic point of view, involving crucial aspects such as the reliability, lifecycle, and environmental sustainability of these materials. In this context, fractional models are becoming more and more popular because of their capability to describe the behavior of such materials using a limited number of parameters. These models can be expressed by adopting a pole–zero formulation. This formulation that in logarithmic scale becomes a superposition of pole–zero couple behavior allows us to overcome the computational and convergence issues that occur in parameter identification of a rheological model in the time domain and frequency domain form.

In this work, first, the validity of the WLF principle through the material transformation to the poles–zeros formulation was demonstrated. After that, a constrained pole–zero identification procedure was used not only to identify both the master curve but also the material WLF function. An extensive study was carried out on three different compounds to evaluate the capability of the FDGM in poles–zeros formulation to reconstruct the master curve in the frequency domain and correctly identify the coefficients of the WLF function, making use of the data concerning the individual isothermal experimental curves from a DMA test. The analysis shows that the fractional calculus in poles–zeros formulation is a powerful analytical model able to reproduce the material viscoelastic behavior, including the time–temperature superposition phenomenon. The FDGM ensures not only a high correlation with the viscoelastic master curves but is also able to estimate the differences in the WLF law of each polymer compound.

As an extension, the paper points out the further possibility of employing only a limited number of the experimental isothermal curves to feed the identification methodology, as the samples specifically concern the material glass transition zone. 

These results highlight how the proposed methodology is able to give a good estimation of the master curves and the time–temperature superposition parameters while using a reduced dataset of experimental data. These results could open favorable scenarios both from an economic and environmental point of view: reducing the number of tests necessary for the preliminary characterization of materials, reducing development and design costs, and increasing predictive knowledge about the behavior of the material during its entire life for reuse and recycling.

## Figures and Tables

**Figure 1 polymers-14-04412-f001:**
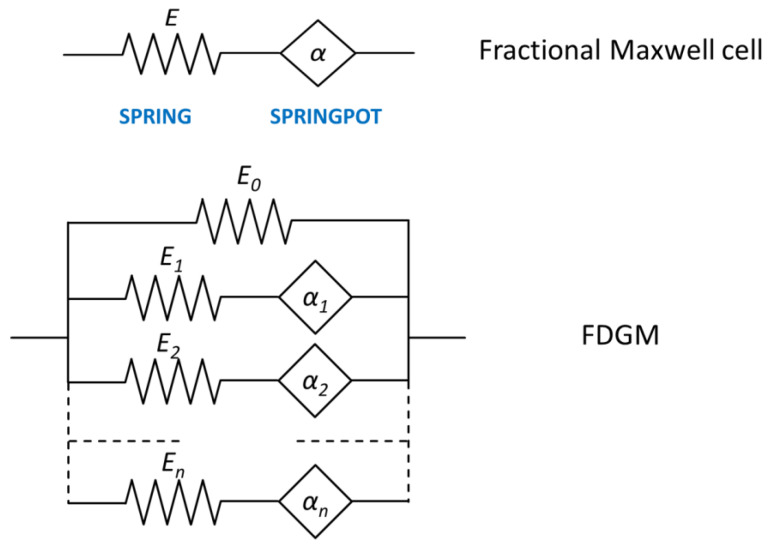
Scheme of the fractional derivative generalized model.

**Figure 2 polymers-14-04412-f002:**
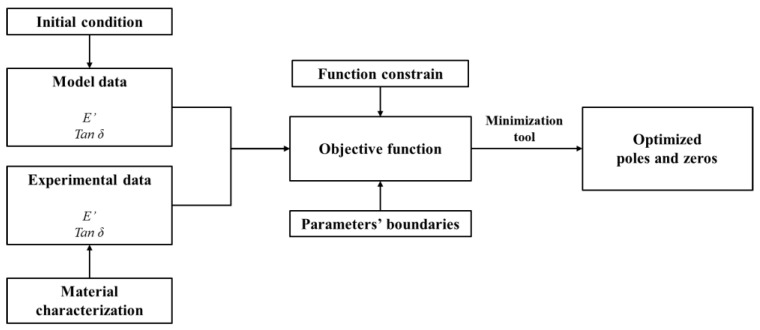
Constrained nonlinear optimization procedure to identify poles–zeros coefficients: functional scheme.

**Figure 3 polymers-14-04412-f003:**
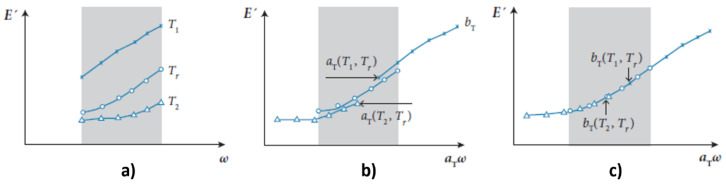
Determination of the frequency–temperature superposition shifting factor *a_T_* and *b_T_*: (**a**) Isotherms of storage modulus on the frequency range measurable by DMA, at temperatures *T*_1_, *T*_2_, and *T_r_*, with *T*_1_ < *T_r_* < *T*_2_; (**b**) Isotherms of storage modulus after application of the horizontal shift factors, taking *T_r_* as the reference temperature; (**c**) Isotherms of storage modulus after application of both horizontal and vertical shift factors, taking *T_r_* as the reference temperature [28].

**Figure 4 polymers-14-04412-f004:**
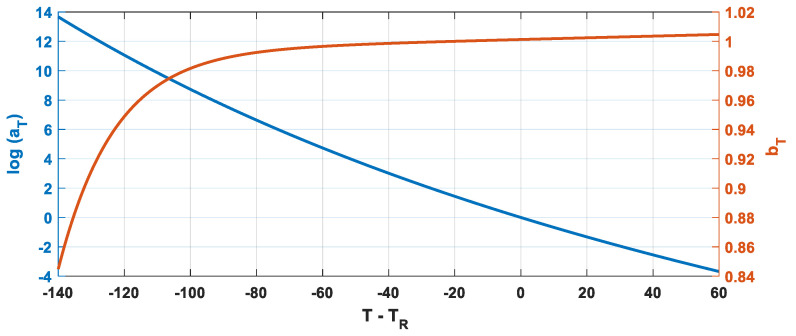
Typical trend of a_T_ and b_T_.

**Figure 5 polymers-14-04412-f005:**
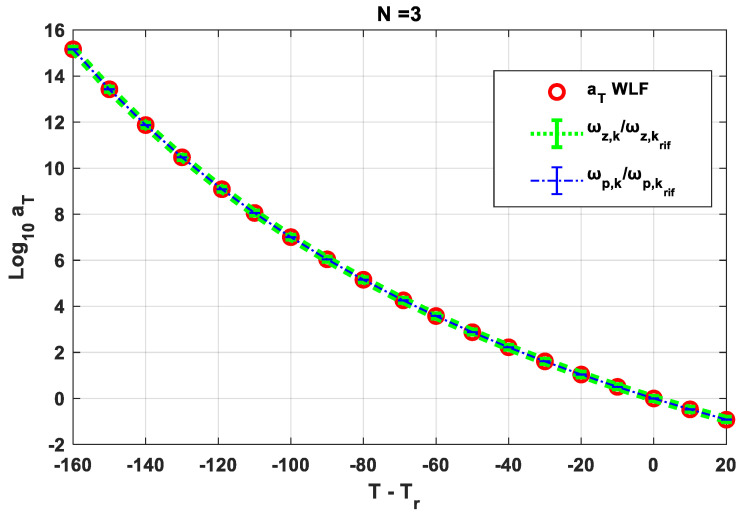
Relation between poles, zeros, and WLF law.

**Figure 6 polymers-14-04412-f006:**
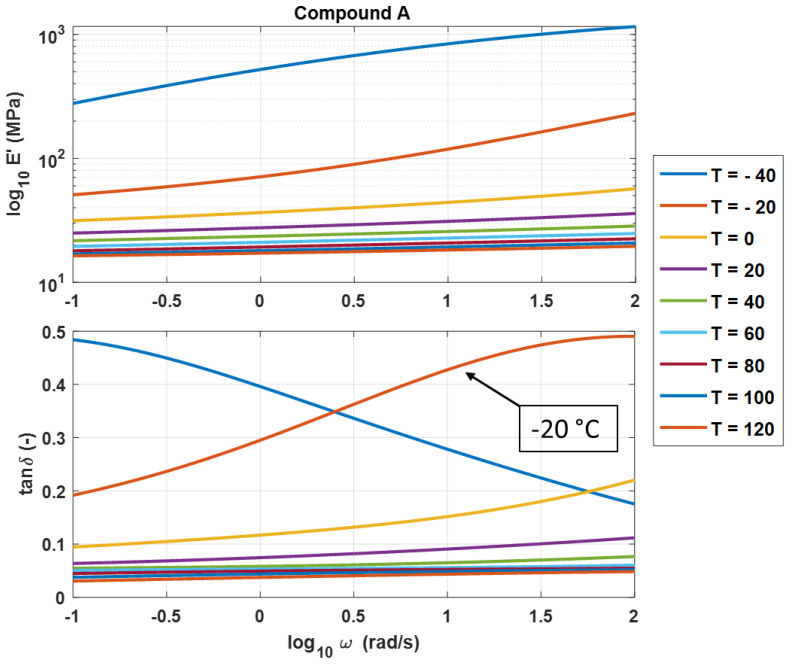
Experimental DMA isothermal curves.

**Figure 7 polymers-14-04412-f007:**
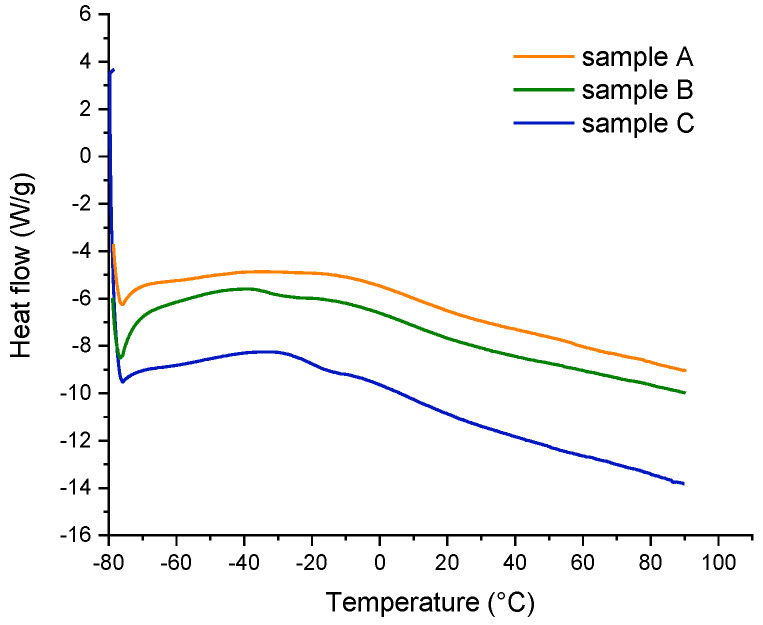
Differential scanning calorimeter testing routine.

**Figure 8 polymers-14-04412-f008:**
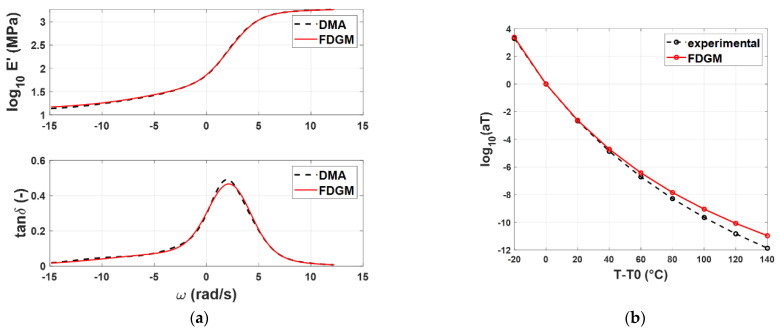
Compound A: Storage modulus and loss tangent (**a**), WLF law (**b**)—experimental data (dashed line) vs. FDGM model.

**Figure 9 polymers-14-04412-f009:**
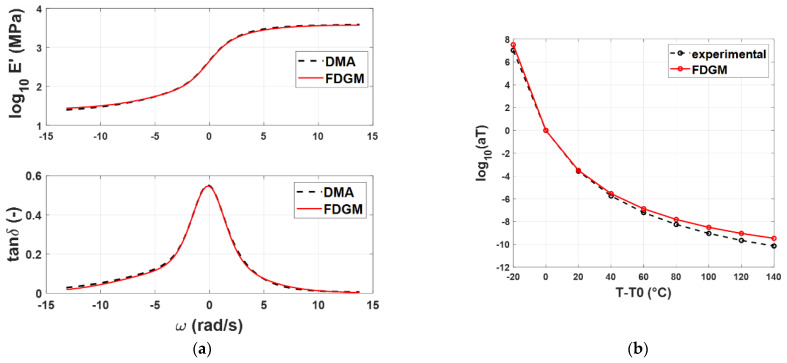
Compound B: Storage modulus and loss tangent (**a**), WLF law (**b**)—experimental data (dashed line) vs. FDGM model.

**Figure 10 polymers-14-04412-f010:**
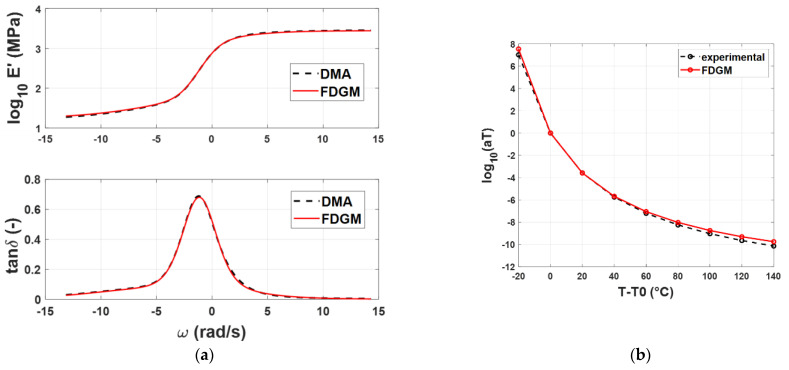
Compound C: Storage modulus and loss tangent (**a**), WLF law (**b**)—experimental data (dashed line) vs. FDGM model.

**Figure 11 polymers-14-04412-f011:**
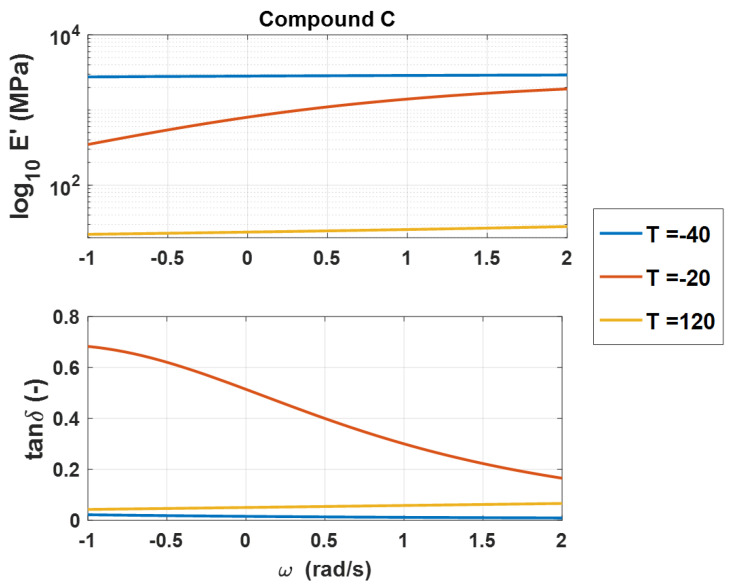
Example of the experimental starting set for the identification with partial DMA for compound C.

**Figure 12 polymers-14-04412-f012:**
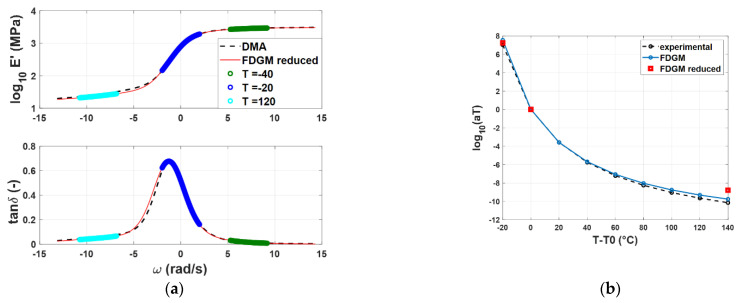
Compound C: Storage modulus and loss tangent (**a**), WLF law (**b**)—experimental data (dashed line) vs. FDGM model complete (light blue) and FDGM obtained with partial staring experimental dataset (red).

**Table 1 polymers-14-04412-t001:** Experimentally identified glass transition temperatures for compounds A, B, and C.

	I T_g_ (°C)	II T_g_ (°C)
Compound A	-	9
Compound B	−33	8
Compound C	−20	9

**Table 2 polymers-14-04412-t002:** NRMSEs and identified WLF coefficients for compounds A, B, and C.

	NRMSE E′ (MPa)	NRMSE tanδ (-)	NRMSE Total	*T_rif_* (°C)	*C_1_*	*C_2_*
Compound A	0.0108	0.0637	0.0372	−20	23.36	158.21
Compound B	0.0486	0.0442	0.0464	−20	13.19	55.12
Compound C	0.0402	0.0406	0.0404	−20	17.31	56.24

**Table 3 polymers-14-04412-t003:** NRMSEs and identified WLF coefficients for compounds A, B, and C in case of partial DMA data.

	NRMSE E′ (MPa)	NRMSE tanδ (-)	NRMSE Total	*T_rif_* (°C)	*C_1_*	*C_2_*
Compound A	0.0339	0.0568	0.0454	−20	30.24	203.53
Compound B	0.012	0.057	0.0345	−20	16.13	65.04
Compound C	0.011	0.11	0.0617	−20	12.15	53.64

## Data Availability

Not applicable.

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
