# Peer review of "Fractional Calculus Approach to Reproduce Material Viscoelastic Behavior, including the Time–Temperature Superposition Phenomenon"

_polymers, 2022, doi:10.3390/polym14204412_

Round 1
Reviewer 1 Report
The present work is a continuation of an Authors' previous work where fractional models coupled to the poles-zeros identification methodology were used to identify viscoelastic master curves. Now the whole same procedure is applied to obtain the material parameters further characterizing the time-temperature superposition principle such as calculating the coefficients of the WLF equation. Moreover, the possibility to optimize the minimum number of the experimental curves needed to the model is also presented. The entire manuscript is well presented and both data results and commentary are overall satisfactory. Here are some constructive observations to help the Authors to improve the interest impact of their work:- One of the important disclaimer by the Authors is that vertical shifts, which take into accounts for thermal expansion contribution of the polymer matrix, are not included in the present analysis. However, in the brief description and schematic representation of the reduced variables procedure reported at pg. 5 it is worth specifying that "thermal expansion vertical shifts can be easily neglected in the viscoelastic regions where the frequency/time dependence of material functions is pretty sharp. On the other hand, overlooking thermal vertical shifts in viscoelastic regions having weak frequency/time dependence may lead to different values of horizontal shifting whose accuracy depends on the examined material";
- From the observation pointed out earlier, it follows that the reduced variables procedure shown in Figure 3 should have depicted the way of shifting the other way around: First thermal vertical shifts, Secondly horizontal shifts;
- In order to appreciate more the potential of the proposed model approach I strongly recommend to give a little more molecular structure information about the differences of sample A, B, and C. From the WLF curves I assumed that B and C have a very similar microstructure with close glass transition temperature Tg. Knowing the Tg can help better test the accuracy of the WLF coefficients obtained experimentally and via modeling by changing the reference temperature;
- From the previous comment, it follows that reporting both experimental and modeling calculated WLF coefficients at three different reference temperatures, for example, may give to the reader a more complete understanding on how reliable the model is for predicting time-temperature superposition phenomenon;
- This might be a simple typo, but the reference temperature Trif = -120C reported in Table 1 sounds unfeasible. Please clarify.
Author Response
Reviewer #1
The present work is a continuation of an Authors' previous work where fractional models coupled to the poles-zeros identification methodology were used to identify viscoelastic master curves. Now the whole same procedure is applied to obtain the material parameters further characterizing the time-temperature superposition principle such as calculating the coefficients of the WLF equation. Moreover, the possibility to optimize the minimum number of the experimental curves needed to the model is also presented. The entire manuscript is well presented and both data results and commentary are overall satisfactory. Here are some constructive observations to help the Authors to improve the interest impact of their work:
- One of the important disclaimer by the Authors is that vertical shifts, which take into accounts for thermal expansion contribution of the polymer matrix, are not included in the present analysis. However, in the brief description and schematic representation of the reduced variables procedure reported at pg. 5 it is worth specifying that "thermal expansion vertical shifts can be easily neglected in the viscoelastic regions where the frequency/time dependence of material functions is pretty sharp. On the other hand, overlooking thermal vertical shifts in viscoelastic regions having weak frequency/time dependence may lead to different values of horizontal shifting whose accuracy depends on the examined material";
The authors agree with the reviewer comment and have modified the paper accordingly. More in detail, at page 5 the following text has been added:
“Here it worth noting that this hypothesis is highly acceptable in the viscoelastic regions where the frequency/time dependence of material functions is sharp. On the other hand, overlooking thermal vertical shifts in viscoelastic regions having weak frequency/time dependence may lead to different values of horizontal shifting whose accuracy depends on the material under investigation.”
- From the observation pointed out earlier, it follows that the reduced variables procedure shown in Figure 3 should have depicted the way of shifting the other way around: First thermal vertical shifts, Secondly horizontal shifts;
Please note that Figure 3 does not refer to the reduced variable procedure. This figure has been introduced to qualitative remind the redear about the determination of the frequency-temperature superposition shifting factor aT and bT and refers to the cited reference, [28]. The text has been modified to point out this explanatory purpose of the figure.
- In order to appreciate more the potential of the proposed model approach I strongly recommend to give a little more molecular structure information about the differences of sample A, B, and C. From the WLF curves I assumed that B and C have a very similar microstructure with close glass transition temperature Tg. Knowing the Tg can help better test the accuracy of the WLF coefficients obtained experimentally and via modeling by changing the reference temperature;
The molecular composition cannot be disclosed due to confidentiality agreements with the material supplier. On the other hand, the authors want to underline that this methodology finds its full utility and innovation when there is no information on the polymer under investigation. If further chemical-mechanical data were available, one could think of integrating the proposed methodology, but this is out of the scope of this work.
- From the previous comment, it follows that reporting both experimental and modeling calculated WLF coefficients at three different reference temperatures, for example, may give to the reader a more complete understanding on how reliable the model is for predicting time-temperature superposition phenomenon;
The choice of the reference temperature is an important part of the proposed procedure. In order to minimize the randomness of the procedure and, therefore, subject the different materials to the same algorithm (compatible with the assumptions made), a unique way to establish the reference temperature has been defined. In particular, as reported in the manuscript, has been assumed as Trif the temperature of the isotherm curve that presents the maximum value of the tan δ (i.e. for the compound A Trif = -20 °C). The manuscript has been modified, at page 8 of the revised version, to point out this concept.
Other techniques for the choice of the Trif, starting from the experimental data could be explored, but it would be necessary to investigate and report in ad-hoc paper. On the other hand, the authors wanted to highlight the reliability of the procedure and the model, applying the methodology on three totally different compounds.
This might be a simple typo, but the reference temperature Trif = -120C reported in Table 1 sounds unfeasible. Please clarify.
Yes, Is a typo that has been fixed. Trif = -20C. Many thanks

Reviewer 2 Report
The paper presented a pole-zero identification methodology that can be used not only to identify the viscoelastic master curves, but also the material parameters that characterize the time-temperature superposition phenomenon.
The proposed technique uses the generalized fractional derivative model to reconstruct the master curves in the frequency domain and correctly identify the coefficients of the WLF function.
The paper is well organized and written, making it easy to read.
The most recent bibliography is from 2020, the others are more than 5 years old. It would be important to see the latest state of the art.
Another suggestion is that the authors make it clear whether the computational results were developed by the authors themselves.
Author Response
Reviewer #2
The paper presented a pole-zero identification methodology that can be used not only to identify the viscoelastic master curves, but also the material parameters that characterize the time-temperature superposition phenomenon.
The proposed technique uses the generalized fractional derivative model to reconstruct the master curves in the frequency domain and correctly identify the coefficients of the WLF function.
The paper is well organized and written, making it easy to read.
The most recent bibliography is from 2020, the others are more than 5 years old. It would be important to see the latest state of the art.
Another suggestion is that the authors make it clear whether the computational results were developed by the authors themselves.
The authors deeply thank the anonymous Reviewer for the positive appreciation of the paper and for his suggestions. The state of art has been integrated adding the latest scientific founding. More in detail the following citation has been added:
“Several applications take advantage of the fractional models. A review, regarding the application of fractional calculus in the models of linear viscoelasticity utilized in dy-namic problems of mechanics of solids has been conducted by Shitikova [13]. Abouelregal [14] proposed a methodology to study thermoelastic vibrations in a ho-mogeneous isotropic three-dimensional solid based a fractional derivative Kelvin–Voigt model. In [15], Zhou et al. adopted a variable-order fractional derivative materi-al model to numerically analyze the behavior of the frozen soil, including creep, stress relaxation and strain rate effects. In [16], Wang et al. adopted the fractional derivative model to describe the hysteretic behavior of the magnetorheological elastomers, demonstrating a huge potential in the field of intelligent structures and devices. “
- Shitikova, M. V. Fractional Operator Viscoelastic Models in Dynamic Problems of Mechanics of Solids: A Review. Mech. Solids 2022, 57.
- Abouelregal, A.E. Thermo-viscoelastic properties in a non-simple three-dimensional material based on fractional derivative Kelvin–Voigt model. Indian J. Phys. 2022, 96.
- Zhou, F. xi; Wang, L. ye; Liu, Z. yi; Zhao, W. cang A viscoelastic-viscoplastic mechanical model of time-dependent materials based on variable-order fractional derivative. Mech. Time-Dependent Mater. 2021.
- Wang, P.; Yang, S.; Liu, Y.; Zhao, Y. Experimental Study and Fractional Derivative Model Prediction for Dynamic Viscoelasticity of Magnetorheological Elastomers. 2022, 1, 3.
The authors would like to point out all computational results were developed by themselves. The paper has been modified accordingly to underline this aspect.

Round 2
Reviewer 1 Report
Finding an "universal" way to capture the TTS phenomenon with the prediction of WLF coefficients has been appealing for many and in that sense a kind of universality has been reached by estimating the WLF coefficients at the glass transition temperature Tg. I do understand that certain info, such as Tg, may be very sensitive to suppliers since it is possible to guess what formulation has been made, but I think that a little mention on what the difference is among sample A, B, and C (for instance, different Tg, Mw, macrostructure, etc) without telling the actual values, would be highly appreciated by the reader to better understand how different the samples are in reality. Moreover, if the Authors confirm that the proposed algorithm estimates values of C1 and C2 at Tg of each sample close to those reported in Ferry's book (17.44 and 51.6, respectively) without disclosing the actual Tg, the reliability of their method would much benefit, since the main issue with the TTS principle and the WLF equation is that both of them are 100% empirical approaches and their results must be compared to previous experiments with similar systems in order to appreciate the results goodness as pointed out by Ferry himself. Furthermore, the proposed reference T at the maximum tanδ may not suit all the polymer candidates (thinking of rubbers crosslinked at low and intermediate level). Thus, the Authors should give a second T_ref option, Tg is just an example, to make their whole methodology of interest for the soft matter community.Author Response
Please see the attachment.

Round 3
Reviewer 1 Report
Following Authors' statement of "while for initial C1 and C2 in (10) they have been assumed 17.44 and 51.60 respectively, that are generally accepted values when T_rif = Tg" at the end of 4.1 paragraph, my question is about to understand if the same calculation procedure can be carried out by simply imposing T_rif = Tg (which is now shown by the Authors to be almost the same for all the three samples, Tg_average ~9C) with other initial values of C1 and C2, and to see if the proposed methodology returns values of C1 and C2 close to 17.44 and 51.60, respectively. Aside the interdependence between Tg and tanδ peak (which is the unique criterion chosen by the Authors) in many polymeric systems, the suggested assessment is of high importance when dealing with master curves and the estimation of WLF coefficients which are often recommended to be tested at both Tg and any other random T as reference since they are empirical approached. Indeed, it is very interesting to observe that for samples having a very similar Tg, sample A shows a different T dependence with respect to sample B and C on on hand, while the Authors' methodology is giving values of C1 and C2 close to those a Trif = Tg but when Trif = Tg-30K for sample B and C on the other. These Authors' findings make it even more interesting to show the double check with Trif=Tg.Author Response
Dear Reviewer,
firstly, the authors would like highlight once again that the paper focuses on the mechanical modeling of viscoelastic material. More in detail, as clearly reported also in the abstract, “the aim of the presented work is to demonstrate that the poles-zeros identification methodology can be employed not only to identify the viscoelastic master curves, but also the material parameters characterizing the time-temperature superposition phenomenon. The proposed technique, starting from the data concerning the isothermal experimental curves, makes use of the fractional derivative generalized model to reconstruct the master curves in the frequency domain and to correctly identify the coefficients of the WLF function.”
In the authors’ opinion having demonstrated this, which took months of studies and research, is an important discovery that is worth sharing with the scientific community. Moreover, the authors want to underline that this methodology finds its full utility and innovation when there is no information on the polymer under investigation. If further chemical-mechanical data were available, one could think of integrating the proposed methodology, but this is totally out of the scope of this work.
As concerns the reviewer question regardinga further investigation “if the same calculation procedure can be carried out by imposing T_rif = Tg (which is now shown by the Authors to be almost the same for all the three samples, Tg_average ~9C) with other initial values of C1 and C2, and to see if the proposed methodology returns values of C1 and C2 close to 17.44 and 51.60, respectively” points out that the purpose of the paper is not clear to the reviewer. Starting from the isothermal curves (reported in fig 6.) the author totally defined criteria to calculate, through the fractional derivative model in pole-zero formulation , also the WLF coefficients.
To the above purpose, the presented procedure, at this stage, is not addressing the knowledge of the Tg temperature in advance.
Following the indications of the previous round of reviews, the authors have performed the additional DSC test as requested by the reviewer to point out the comparisons between them not only from viscoelastic curves point of view and to give the reader more useful information.
Observing Figure 6, is clear that there are no isothermal curves at T= Tg measured by DSC. Imposing Tg =Trif is out of the scope of this paper and it would imply having to do the DMA tests again. This may be for sure an interesting evolution of this work, but it will also represent another study that could be addressed in another specific paper.
For these reasons the authors disagree with the revisions and ask for the manuscript to be published in its current form.